# Toward the Required Detection Limits for Volatile Organic Constituents in Marine Environments with Infrared Evanescent Field Chemical Sensors

**DOI:** 10.3390/s19173644

**Published:** 2019-08-21

**Authors:** Carina Dettenrieder, Yosef Raichlin, Abraham Katzir, Boris Mizaikoff

**Affiliations:** 1Institute of Analytical and Bioanalytical Chemistry, Ulm University, 89081 Ulm, Germany; 2Department of Physics, Faculty of Natural Sciences, Ariel University, Ariel 44837, Israel; 3School of Physics and Astronomy, Sackler Faculty of Exact Sciences, Tel Aviv University, Tel Aviv 69978, Israel; 4Department of Chemical Sciences, University of Johannesburg, Johannesburg 2001, South Africa

**Keywords:** mid-infrared, fiber-optic evanescent field sensor, optical chemical sensor, chemosensor, IR sensor, Fourier transform infrared spectroscopy, silver halide fiber, polymer coating, enrichment, environmental monitoring, volatile organic compound, VOC

## Abstract

A portable sensor system for the simultaneous detection of multiple environmentally relevant volatile organic compounds (VOCs) in real seawater based on Fourier transform infrared fiber-optic evanescent wave spectroscopy (FT-IR-FEWS) was developed. A cylindrical silver halide (AgX) fiber with an ethylene/propylene copolymer (E/P-co) coated flattened segment was used as an active optical transducer. The polymer membrane enriches the hydrophobic analytes, while water is effectively excluded from the penetration depth of the evanescent field. Determination of multicomponent mixtures (i.e., 10 VOCs in real-world seawater samples) collected in Arcachon Bay, France revealed a high accuracy and reproducibility with detection limits down to 560 ppb. The measurement showed no significant influence from changing water conditions (e.g., salinity, turbidity, and temperature or other interfering substances). The time constants for 90% saturation of the polymer ranged from 20 to 60 min. The sensor system is capable of being transported for on-site monitoring of environmental pollutants in aqueous matrices with efficient long-term stability, thus showing great potential to be utilized as an early warning system.

## 1. Introduction

Increasing industrialization and population of coastal regions and the use of the ocean as a dumping ground for contaminated dredged material, sewage sludge, industrial, and domestic waste has led to increased pollution of the oceans, which is one of the most important environmental problems [1]. In particular, organic contaminants such as volatile organic compounds (VOCs) are the most commonly detected pollutants [2]. VOCs are in part carcinogenic, mutagenic, and toxic, and may damage the health of humans and the marine ecosystem [3]. The widespread use of VOCs in the manufacturing and handling of paints, solvents, adhesives, fuels, pharmaceuticals, deodorants, fumigants, and refrigerants or leaking underground pipelines, solvent storage vessels, landfill, and manufacturing effluents has led to releases into the environment [4,5,6]. More than half of the total number of identified substances originating from sea-based sources results from offshore gas/oil activities (39%) and shipping (32%) [1]. Maritime traffic on the world’s seas, with about 215 million tons of chemicals in 2015, has led to discharges, even though this is illegal [1]. 

Certain VOCs may persist in groundwater and move into drinking water supplies. Therefore, VOCs have been detected in aqueous matrices such as in ground and surface waters [7,8,9,10,11], drinking water [12,13,14,15,16], river water [17,18,19], and seawater [20,21,22,23,24,25,26,27,28]. In general, organic pollutants that might have a toxicological impact have been found in many marine areas (e.g., northwestern Mediterranean Sea, the Adriatic Sea, the harbor of Naples, Haifa, the Saronikos Gulf, the Sea of Marmara, and the northwestern Black Sea) [29,30]. Of all of the substances released into seawater, most contaminants originate from offshore oil and gas operations [1]. For example, as of September 2015, over 528,000 releases from underground petroleum storage tanks and pipes have been reported since the underground storage tank program was implemented [31]. Every year, 10%–15% of the oil entering the oceans, and thus VOCs, are from oil tanker accidents [1]. 

Therefore, the protection of global waters and the reduction of discharges, emissions, and losses of hazardous substances and maintaining public health are of substantial communal, political, legislative, and economic interest for several commissions such as the US Environmental Protection Agency (US EPA) and the European Analytical Quality Control in support of the Water Framework Directive for sustained health of the marine ecosystem.

In order to protect marine oceans and the surrounding ecosystems, monitoring hazardous pollutants in seawater with an adequate method is necessary. Conventionally, the detection of VOCs is performed using high-pressure liquid chromatography (HPLC) or gas chromatography coupled with mass spectrometry (GC-MS), which offers high sensitivity in the low ppb range. For sample preconcentration, purge-and-trap [32,33] or solid-phase microextraction [34,35,36] are mostly used. Furthermore, spectroscopy-based methods such as ultra-violet (UV) [37,38], Raman [39], near infrared [40], and fluorescence [41] spectroscopy have been applied for VOC detection. However, these methods are expensive and time-consuming due the bulky instrumentation required, therefore, the sample is collected discontinuously and transported to a well-equipped laboratory. Sample pretreatment steps (purification and extraction) may lead to erroneous results due to sample losses from the volatile nature of the analytes to be investigated [42]. Therefore, skilled personnel are required during this work. 

In particular, detection in seawater is challenging due to its complex matrix such as high ionic strength and the presence of marine humus, resulting from organic matter like cellular debris, humic acids, and metabolic products as well as other interfering dissolved organic substances and bio-organisms. The diversity of the analytes and the permanent changing concentrations and conditions both spatially and temporally that are related to estuaries or effluents makes detection challenging. Consequently, a rapid in situ and real-time screening sensor for on-site measurement of VOCs in aqueous matrices is envisaged such as infrared attenuated total reflection (IR-ATR) spectroscopy.

IR-ATR spectroscopy has the advantage of being a non-destructive method, and can analyze pollutants in situ and in real-time since sampling and the sample pre-treatment steps are not necessary. In the mid-infrared (MIR) spectral range (3–20 µm), well-structured substance-specific absorption bands are caused, in particular, in the so-called fingerprint region (below 1100 cm^−1^) by the excitation of the fundamental vibrational states of the analyte molecules. 

The principle of ATR is exemplarily illustrated with AgX fiber as an optical transducer in Figure 1. The light is totally internal reflected at both sides of the waveguide at the interface to the surrounding medium (E/P-co), if the incident IR beam is at an angle of incidence $\theta_{i}$ larger than the critical angle $\theta_{c}$. The refractive index of the waveguide $n_{1}$ (i.e., the AgX fiber) has to be larger than the refractive index of the surrounding medium $n_{2}$ (E/P-co polymer coating), as defined by Snell’s law:(1)$$\theta_{C}=\arcsin\frac{n_{2}}{n_{1}}$$

The theoretical number of total internal reflections N within a waveguide can be calculated with the following equation:(2)$$N=\frac{L}{a}=\frac{L}{d\tan\theta_{i}}$$

where L and d are the length of the sensing region and the waveguide thickness, respectively. a is the distance between each reflection. By decreasing the waveguide thickness, the angle of incidence changes, leading to an increased penetration depth and number of total internal reflections, and therefore, an increased sensitivity. Hence, a segment of cylindrical AgX fiber (700 µm) was press tapered to a diameter of 150 µm. 

Due to interference of the incident and reflected light, an evanescent field is generated that penetrates into the polymer layer depending on the wavelength of the incident beam λ, the refractive indices $n_{1}=2.2$ (AgX fiber) and $n_{2}=1.48$ (E/Pco), and the angle of incidence $\theta_{i}=45^{{}^{\circ}}$:(3)$$d_{p}=\frac{\lambda }{2\pi \sqrt{n_{1}^{2}\sin^{2}\theta_{i}-n_{2}^{2}}}$$

The maximum penetration depth at 12.65 µm (790 cm^−1^) was calculated as 4.20 µm. The intensity of the evanescent field decays exponentially with the distance x:(4)$$E\left(x\right)=E_{0}\exp\left(-\frac{x}{d_{p}}\right)$$
where $E_{0}$ is the intensity at the interface. 

Conventionally, chalcogenides glasses, heavy metal fluorides, tellurium halides, and AgX are used as fiber-optic materials [43]. AgX fibers (AgCl_x_Br_1−x_; 0<x<1) are among the most promising materials for MIR sensing applications as they have the advantage of being flexible, and transparent in the entire MIR spectral range, depending on its composition [44]. Cylindrical fibers [45,46,47], flattened fibers [48], and flattened fibers with cylindrical extensions on both ends [49,50,51] have already been investigated. In this study, a planar AgX fiber with cylindrical extension was used since it has been reported to have enhanced absorbance signals when compared to cylindrical fibers while maintaining good properties of incoupling the IR radiation [51].

The surface of the waveguide (i.e., flattened segment) was coated with a polymer membrane acting as a solid-phase extractor to enable the continuous measurement of pollutants over a long period of time. The hydrophobic polymer only enriches hydrophobic analytes (i.e., pollutants), which interact with the generated evanescent field. In MIR spectroscopy, high water absorption bands interfere with the specific absorption features of the analytes. Due to the hydrophobic nature of the polymer membrane, water is effectively excluded from the evanescent field. This enables MIR spectroscopy sensing in aqueous matrices with an increased sensitivity. At the same time, the polymer coating prevents the AgX fiber from being exposed to UV radiation and chloride ions. Otherwise, a water soluble silver complex [AgCl_2_]^−^ is formed, and degradation of the optical fiber will take place [52]. The thickness of the coating layer must exceed the penetration depth of the evanescent field in order to efficiently restrain the water molecules. However, too thick membranes will lead to long enrichment times to obtain steady-state conditions. A wide variety of polymeric materials for the enrichment membrane have already been investigated including polyisobutylene (PIB) [53,54,55], low-density polyethylene (LDPE) [56], E/P-co [49,57], Teflon® AF [58,59], or poly(dimethylsiloxane) [59,60]. Based on the superior enrichment properties due to its amorphous nature, and therefore, its high free volume, the E/P-co polymer was used within this study. The feasibility of MIR-FEWS for pollutant monitoring in aqueous matrices (i.e., water and artificial seawater) through the use of a polymer coated polycrystalline AgX fiber as a sensing element has already been demonstrated by several research teams [47,49,50,61,62,63,64,65]. However, these studies have shown simultaneous detection up to six (chlorinated) hydrocarbons in water using a liquid nitrogen-cooled detector. Due to the liquid nitrogen, this is not applicable for in-field environmental analysis. Kraft et al. [61] have demonstrated the detection of five VOCs in artificial seawater through the use of a Stirling-cooled detector. Salinity, turbidity, inorganic ions, and marine humus do not influence the performance of the sensor. Additionally, they investigated cross interferences ranging from highly hydrophobic (*n*-hexane) to hydrophilic (methanol and pyridine) substances, showing that enrichment factors and sensor response remained constant [66].

The present study describes a field-deployable portable IR spectroscopic sensor system developed during a European Union project (SCHeMA, “Integrated in situ chemical mapping probes”, Grant Agreement Number 614002 [67] toward the required detection limits for continuous in situ and real-time environmental pollution monitoring, especially regarding detection in complex matrices (i.e., real seawater). Therefore, implementation as a “threshold alarm sensor” was envisaged. The MIR sensor system is capable of analyzing simultaneously multiple VOCs (i.e., environmental pollutants). Therefore, a mixture of ten VOCs in real seawater collected from Arcachon Bay, France were successfully qualified and quantified. Detection limits in the low ppm region with a rapid and reversible enrichment were achieved. Therefore, the developed MIR sensor system facilitates a variety of applications including wastewater or effluent monitoring, underground fuel storage tanks and pipeline monitoring, and controlling drinking water supplies directly on-site from the point of intake and discharge.

## 2. Experimental

### 2.1. Materials and Reagents

Tetrachloroethylene (TeCE, ≥99.9%), (+)-3-carene (CAR, 90%), *p*-xylene (pXYL, 99%), ethylbenzene (EB, ≥99%), 1,2-dichlorobenzene (12DCB, ≥98%), 1,2,4-trichlorobenzene (124TCB, ≥98%), and methanol (99.8%) were purchased from Merck KGaA (Darmstadt, Germany). Myrcene (MYR, 90%) and *p*-cymene (pCYM, 99%) were obtained from Acros Organics (Geel, Belgium). E/P-co (60:40) was supplied by Aldrich Chemical Company (Milwaukee, WI, USA). Trichloroethylene (TCE, ≥99.5%) and 1,3-dichlorobenzene (13DCB, 98%) was obtained from Sigma-Aldrich Chemie GmbH (Munich, Germany). *n*-hexane was obtained from Carl Roth GmbH + Co. KG (Karlsruhe, Germany). The chemicals were used without further purification. 

The flattened AgX fiber (AgCl_0.3_Br_0.7_) with cylindrical extensions used as a sensing element was provided by the research team of Abraham Katzir (Tel Aviv University, Tel Aviv, Israel). The dimensions were as follows: flattened section: length of 45 mm, width of 5 mm, and thickness of 150 µm; cylindrical end facets: length: 15 mm and diameter: 0.7 mm. The AgX fiber was located inside a flow cell produced by the machine shop of Ulm University. According to Equation (4), the evanescent field intensity dependent on the distance from the waveguide surface was calculated. Therefore, $E_{0}$ was measured using a thermopile detector (Gentec-EO, Quebec City, QC, Canada). The intensity of the evanescent field at the penetration depth of 4.2 µm (compared to Equation (3)) at each internal reflection was reduced to 37% of $E_{0}$.

### 2.2. Polymer Coating

A total of 0.5 g of the E/P-co polymer was dissolved in *n*-hexane under reflux for 30 min. The hot solution was dip-coated with an Eppendorf pipette onto both sides of the AgX fiber, 80 µL, each. The polymer film produced was kept at room temperature for at least 2 h until the solvent. The cross-section of the polymer-coated (on top and bottom) AgX fiber using scanning electron microscopy is illustrated in Figure 2. A uniform layer was obtained using the dip-coating method with an Eppendorf pipette.

With differential weighing, the membrane thickness was calculated as 13.7 ± 0.2 µm with differential weighing [54]:(5)$$d=\frac{m_{Polymer}}{\rho_{Polymer}\cdot A}$$
where d= the E/P-co thickness, $m_{Polymer}=\;\mathrm{the}$ mass of the polymer (obtained from weighing the AgX fiber before and after the coating process), $\rho_{Polymer}=$ density of the polymer membrane (0.86 g/cm^3^) [56], and A= the coated surface area (2.25 cm^2^). 

Before starting the first measurement, the E/P-co polymer coating was equilibrated to water since a small amount diffuses into the membrane, thus causing a baseline shift. Due to the water absorption, a baseline shift occurs. The water diffusion reached equilibrium after at least 24 h of exposure. 

### 2.3. Sample Preparation

All analytes (TeCE, TCE, MYR, CAR, pXYL, EB, 12DCB, 13DCB, 124TCB, and pCYM) were dissolved in pure methanol in 20 mL headspace vials resulting in a final concentration of 2000 ppm each. The stock solution was freshly produced daily to ensure no evaporation loss. Sample solutions were prepared prior to the measurements. Therefore, a certain quantity of stock solution was dissolved in the seawater, which was directly collected from Arcachon Bay (Arcachon, France). The amount of methanol in the sample solution, which acted as a solubility mediator, was 1% (v/v). As already described by M. Kraft and B. Mizaikoff [66], spectral interferences did not occur.

### 2.4. Instrumentation and Data Processing

A compact Alpha FT-IR spectrometer in the OEM (original equipment manufacturer) version (Bruker, Ettlingen, Germany) was used as the MIR light source. The IR radiation was focused via a gold-coated elliptical mirror (Thorlabs, Dachau, Germany) and an off-axis parabolic mirror (OAPM, focal length: 25.4 mm, Thorlabs, Dachau, Germany) on the cylindrical end facet of the AgX fiber, which acted as the internal reflection element. Two consecutive OAPMs (Thorlabs, focal length: 25.4 mm and 50.8 mm, Dachau, Germany) were used to focus the IR radiation emanating from the AgX fiber onto a thermoelectrically-cooled MCT (mercury cadmium telluride) detector (Vigo, Poland). For system control and data evaluation, an industrial computer (NST GmbH) was used. All components were mounted on top of an optical breadboard. A schematic and optical image of the MIR-FEWS sensor system is shown in Figure 3. The entire sensor system was reduced in size to 650 × 210 mm^2^ and fit inside a stainless-steel housing for transportation for on-site measurements (i.e., in this study, at Arcachon Bay, France). The sample solution was pumped via polytetrafluoroethylene tubes and tubing connectors inside and out of the flow cell using a peristaltic pump (Watson-Marlow, Cornwall, UK) at a flow rate of 3.2 mL/min. For the rinsing process, the flow rate was set to 4.0 mL/min for a faster regeneration of the polymer coating.

Spectra were recorded in the spectral range from 4000 to 790 cm^−1^ with a spectral resolution of 2 cm^−1^, and 50 scans were averaged per spectrum. Clean seawater from Arcachon Bay was used as the background spectrum. A manual baseline correction was applied to the obtained spectra and evaluated via the univariate method (i.e., peak area analysis). The integration limits are listed in Table 1.

## 3. Results and Discussion

### 3.1. Analysis of Pollutants in Seawater

The AgX fiber was coated with the hydrophobic E/P-co polymer (thickness: 13.7 ± 0.2 µm) in order to enrich the hydrophobic pollutants dissolved in seawater. Therefore, molecules within the penetration depth of the evanescent field were simultaneously detected. Figure 4 shows an exemplary IR absorption spectrum of 10 analytes in seawater with a concentration of 25 ppm each, after being enriched into the polymer membrane. The spectrum was recorded after an enrichment time of 80 min. Each analyte featured one or more clearly visible characteristic absorption bands in the fingerprint region due to the molecule-specific C–H and C–Cl stretching and deformation vibrations. The IR peaks were labeled for clarity: MYR (1594 cm^−1^ and 892 cm^−1^), 13DCB (1577 cm^−1^ and 1412 cm^−1^), pCYM (1515 cm^−1^ and 815 cm^−1^), EB (1495 cm^−1^), 124TCB (1457 cm^−1^, 1094 cm^−1^, 1035 cm^−1^, and 868 cm^−1^), 12DCB (1434 cm^−1^), CAR (1384 cm^−1^ and 989 cm^−1^), TCE (931 cm^−1^), TeCE (910 cm^−1^), and pXYL (795 cm^−1^). Therefore, a multivariate data evaluation was not required. The absorption peaks were consistent in the frequency positions when compared to measurements from the literature performed in deionized water [68]. For univariate data analysis, one peak for each analyte was selected (compare Table 2). The IR spectrum of the E/P-co polymer revealed an absorption band at 804 cm^−1^ [54]. This peak overlapped the peak occurring from pCYM at 815 cm^−1^, therefore, the peak at 1515 cm^−1^ was used for evaluation. Spectral interferences from water absorption bands did not occur (i.e., the polymer coating successfully restrains the water from the evanescent field). Furthermore, the complex and corrosive seawater matrix with its variety of organic matter did not influence the IR signature. Macro-molecules (e.g., humic acids) were not enriched into the E/P-co polymer due to steric hindrance. 

### 3.2. Enrichment Process

The partitioning process (i.e., integrated peak area over the enrichment time) of TeCE into the amorphous E/P-co polymer is exemplarily illustrated in Figure 5 for 10 ppm. For all analytes at the respective concentration, similar enrichment functions could be obtained. The $t_{90}$-value (i.e., the time until 90% of saturation was achieved), ranged from 3 to 60 min. However, the $t_{90}$-value of most of the analytes was around 30 min.

### 3.3. Regeneration

The polymer membrane can be regenerated by flushing a mixture of methanol and clean seawater (20:80, *v*/*v*) through the flow cell. In order to achieve fast regeneration, the flow rate was set to maximum speed (i.e., 4 mL/min). The regeneration process was performed until the peak area decreased to the initial peak area. The diffusion out of the polymer layer over the time is exemplarily shown for TCE in Figure 6. The membrane was fully regenerated after approx. 20 min. Depending on the type of analyte, the regeneration process was longer (50–80 min). The methanol fraction in the rinsing solution did not destroy the E/P-co polymer layer. In order to reduce the time needed for regeneration, the methanol fraction in the mixture can be increased. Therefore, the hydrophobic polymer could be used for multiple measurement cycles, thus maintaining the enrichment properties and protection of the AgX fiber against the corrosive chloride ions contained in seawater. 

### 3.4. Sensor Calibration

In Figure 7, the established calibration functions of all simultaneously detected VOCs in seawater obtained with the developed IR evanescent field sensor are shown. To achieve optimum accuracy, the data were evaluated using the equilibrium method. Therefore, the equilibrium concentration was used to obtain the calibration functions. In order to ensure steady-state conditions were achieved, 10 repetitive measurements were executed after 80 min of enrichment time. The concentration for the respective analyte ranged from 1 ppm to 25 ppm. Calibration functions were obtained by plotting the peak area versus the respective concentration and applying a linear fit.

All obtained linear fits resulted in $r^{2}$-values (goodness of the fit) greater than 0.97. For 12DCB, 124TCB, and CAR, the $r^{2}$-value was even higher than 0.999. The limit of detection (LOD) and limit of quantification (LOQ) were calculated according to the International Union of Pure and Applied Chemistry (IUPAC) through the 3σ- or 10σ-criteria, respectively (3- or 10-times the standard deviation of the smallest concentration value measured). Table 2 summarizes the calibration parameters, LOD, and LOQ for each analyte. Ten VOCs were successfully measured qualitatively and quantitatively in seawater. This is the first time that ten VOCs were simultaneously detected with the resulting detection limits in the low-ppm to ppb concentration range by applying only 50 averaged scans. The quantification of pXYL might be unreliable due to the cut-off wavelength of the thermoelectric MCT detector at 793 cm^−1^, however, pXYL can be determined qualitatively. The $r^{2}$-value, LOD, and LOQ can be further decreased by increasing the number of averaged scans. However, the main purpose of the developed MIR sensor is the rapid detection of VOCs in seawater (i.e., threshold alarm sensor), hence the number of averaged scans for measurements was reduced, thus reducing the time for a single measurement. The use of a liquid nitrogen-cooled MCT detector would lead to enhanced sensitivity as already demonstrated by several research teams [49,50,56,57,63]. However, liquid nitrogen is not applicable for environmental analysis. Changes in salinity (28–34 PSU), turbidity (2.2–14 NTU), temperature (17–22 °C), and pH (7.8–8) over the measurement period revealed no significant influence on the sensor performance, as already described by Mizaikoff [69]. 

### 3.5. Reproducibility

The reproducibility of the developed MIR sensor system was investigated through five repetitive samples. Each sample was prepared successively. The standard deviation of the respective measurement number represents 10 consecutive measurements after the diffusion equilibrium has been achieved. Based on the five independent measurements, the mean peak area and standard deviations were determined. The mean peak area values vs. the number of measurements (i.e., sample number) are shown for each VOC in Figure 8. 

## 4. Conclusions

A sensor for the simultaneous detection of multiple VOCs in a real seawater matrix based on MIR evanescent field spectroscopy was presented. The developed sensor comprised a compact FT-IR spectrometer, coupling optics, and a thermoelectrically-cooled MCT detector. A AgX fiber, located inside a µ-flow cell, was used as a waveguide. The actively transducing part (i.e., flattened segment) was coated with an E/P-co polymer. This hydrophobic membrane effectively excluded the water from the penetration depth of the evanescent field, therefore, the seawater matrix did not influence the absorption features of the respective VOC. Furthermore, protection of the AgX fiber was preserved. A mixture containing 10 different VOCs in seawater, which had been sampled from the coastal region of Arcachon, France, were successfully enriched into the E/P-co polymer layer. Linear calibration functions were established with detection limits in the low ppm to ppb concentration range, which were comparable to the literature based on the low number of averaged scans applied during the individual measurements. It has been demonstrated that the MIR chemical sensor showed reversible enrichment behavior due to the recovery of the polymer membrane by flushing a methanol/water mixture through the flow cell. Hence, multiple measurement cycles over a long-time period are guaranteed. Five repetitive measurements of the same concentration showed good reproducibility. During the measuring period, changing seawater conditions (i.e., salinity, turbidity, temperature, pH, and other unidentified organic matters) did not significantly affect the performance of the sensor. The entire sensor system was miniaturized in size and fit inside a transportation housing, thus enabling direct analysis at the point of discharge. In contrast to conventionally used techniques such as GC-MS or HPLC, the sensor is capable of being deployed on-boat or on-buoy. Future integration into a submersible housing for application in deep sea can be enabled, therefore, the MIR sensor is applicable for threshold alarm sensing (i.e., on-line wastewater or effluent monitoring or controlling underground fuel storage tanks, pipelines, producing oil platforms, and deep-water dumpsites) for the protection of global waters and the marine ecosystem. For drinking water monitoring, the LOD has to be further decreased to achieve the guidelines set by the European Union and other commissions (US EPA, the World Health Organization). Therefore, the number of averaged scans or the number of total internal reflections (i.e., new waveguide design) should be increased. Further improvements can be achieved by optimizing the flow cell design. 

## Figures and Tables

**Figure 1 sensors-19-03644-f001:**
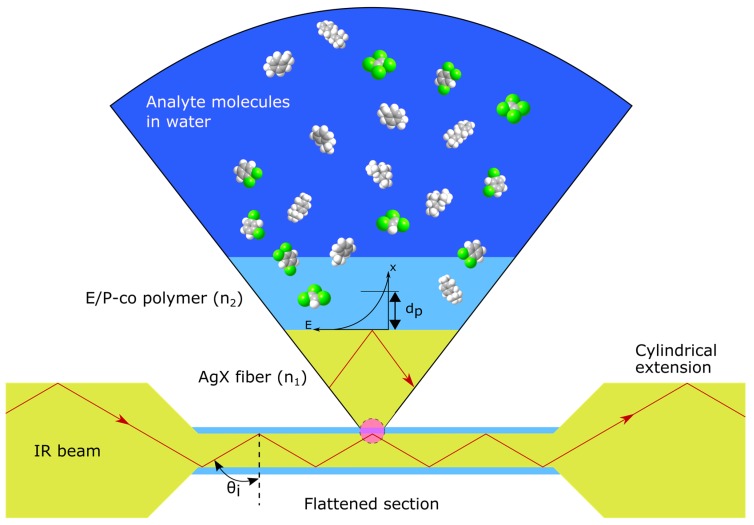
Schematic attenuated total reflection principle of an E/P-co polymer coated AgX fiber. The number of total internal reflections is increased within the flattened section of the AgX fiber. Analyte molecules in an aqueous solution (dark blue) enrich into the polymer layer (light blue) and are detected within the penetration depth $d_{p}$ of the exponentially decaying evanescent field. $\theta_{i}$ is the angle of incident light, $n_{1}$ and $n_{2}$ are the refractive indices of the AgX fiber and E/P-co polymer, respectively, E is the intensity of the evanescent field depending on the distance x.

**Figure 2 sensors-19-03644-f002:**
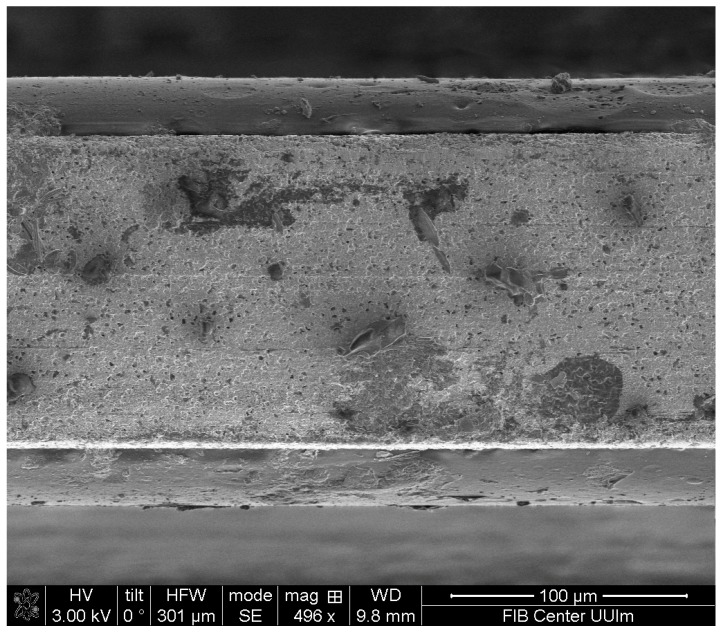
Scanning electron microscopy image of the cross-section of a AgX fiber with the E/P-co polymer coating on the top and bottom revealing a uniform layer.

**Figure 3 sensors-19-03644-f003:**
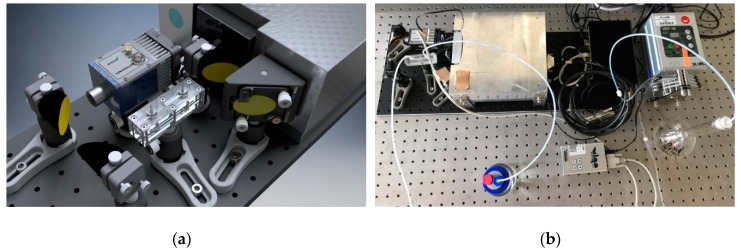
(**a**) Schematic of the experimental setup showing the respective components, FT-IR (Fourier transform infrared) spectrometer, coupling optics, thermoelectrically-cooled MCT detector, and µ-flow cell. The latter included the waveguide element (i.e., AgX fiber). (**b**) Entire miniaturized MIR sensor system (size: 650 × 210 mm^2^) comprising of the optical components, a small computer for data generation and analysis, and a peristaltic pump.

**Figure 4 sensors-19-03644-f004:**
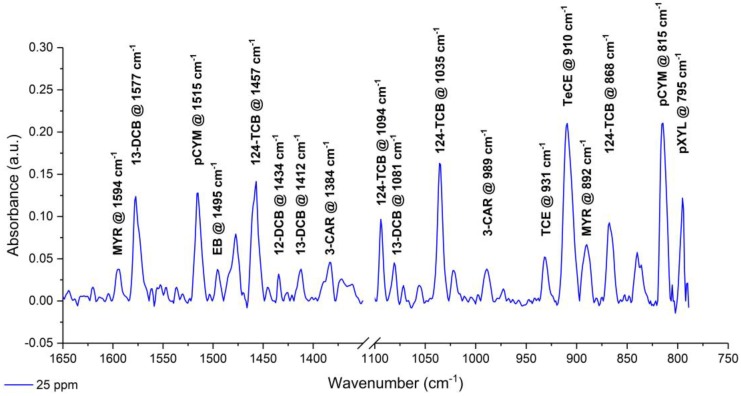
Infrared (IR) absorption spectrum of 10 VOCs (i.e., MYR, 13DCB, pCYM, EB, 124TCB, 12DCB, CAR, TCE, TeCE, and pXYL) in seawater measured simultaneously in a concentration of 25 ppm each. The spectrum was recorded after 80 min of enrichment.

**Figure 5 sensors-19-03644-f005:**
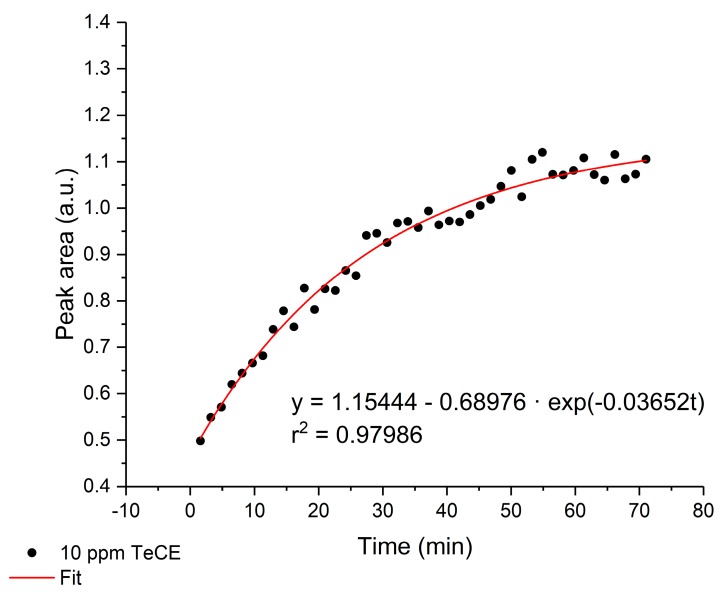
Exemplary diffusion curve of TeCE at a concentration of 10 ppm in an E/P-co polymer layer at 910 cm^−1^. The enrichment process was fitted with a limited growth function.

**Figure 6 sensors-19-03644-f006:**
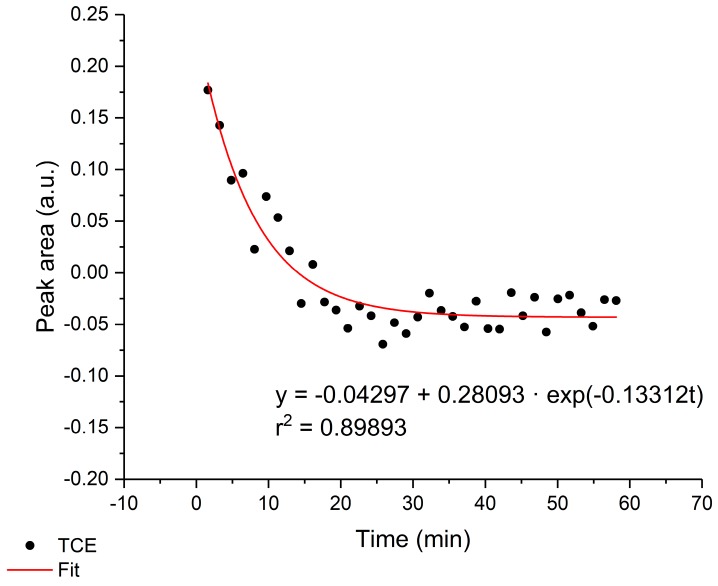
Exemplary regeneration process of TCE at 931 cm^−1^. The polymer layer was completely regenerated after approx. 20 min.

**Figure 7 sensors-19-03644-f007:**
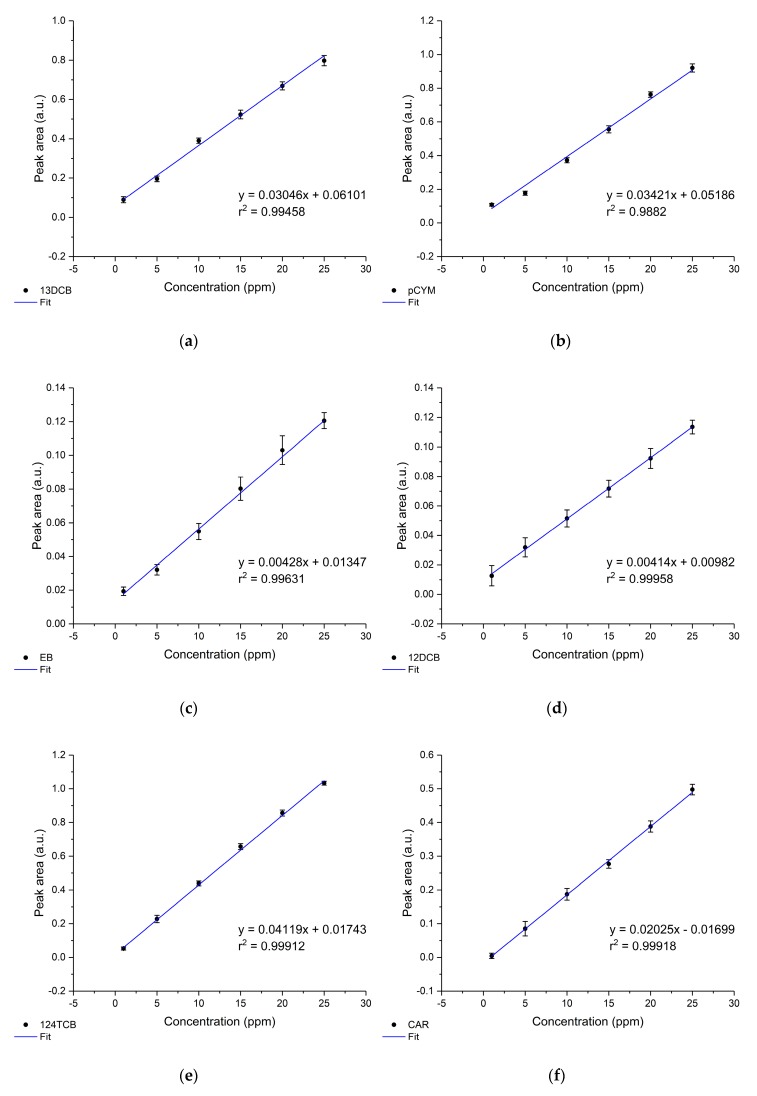
Established calibration functions of the simultaneously determined volatile organic compounds (VOCs) in seawater by the developed MIR chemical sensor. The concentration ranged from 1 to 25 ppm. Data points were obtained by peak area integration. (**a**) 13DCB, (**b**) pCYM, (**c**) EB, (**d**) 12DCB, (**e**) 124TCB, (**f**) CAR, (**g**) TCE, (**h**) TeCE, (**i**) MYR, and (**j**) pXYL.

**Figure 8 sensors-19-03644-f008:**
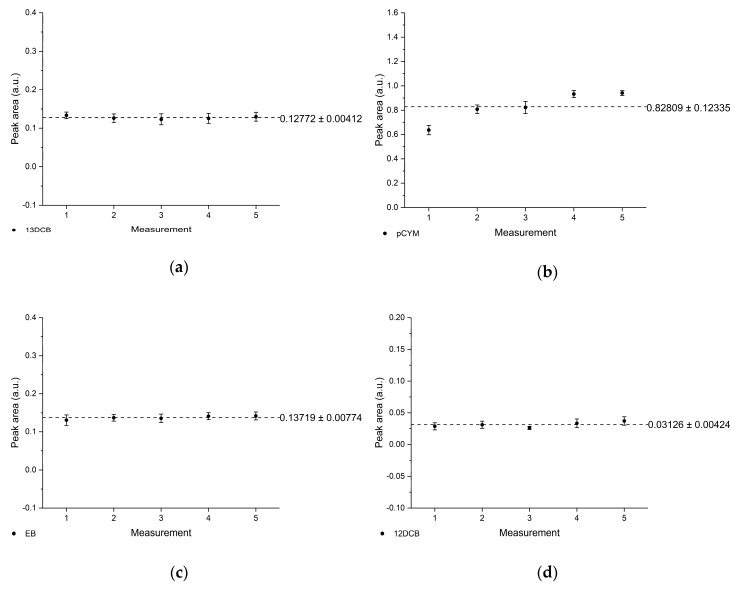
Reproducibility of the MIR chemical sensor system determined from five independent sample measurements demonstrated for (**a**) 13DCB, (**b**) pCYM, (**c**) EB, (**d**) 12DCB, (**e**) 124TCB, (**f**) CAR, (**g**) TCE, (**h**) TeCE, (**i**) MYR, and (**j**) pXYL.

**Table 1 sensors-19-03644-t001:** Integration limits used for the univariate data evaluation (i.e., peak area analysis).

Substance	Peak Area (cm^−1^)
13DCB	1583–1574
pCYM	1521–1506
EB	1498–1489
12DCB	1437–1431
124TCB	104–1028
CAR	1002–981
TCE	939–926
TeCE	915–899
MYR	898–883
pXYL	803–793

**Table 2 sensors-19-03644-t002:** Summary of calibration functions (i.e., linear fit, $r^{2}$-value, LOD and LOQ derived from peak area analysis) over a concentration range from 1 ppm to 25 ppm.

VOC	Wavenumber (cm^−1^)	Linear Fit	$r^{2}$	LOD (ppm)	LOQ (ppm)
13DCB	1577	0.03046x+0.06101	0.99458	1.385±0.030	4.618±0.101
pCYM	1515	0.03421x+0.05186	0.9882	0.669±0.011	2.229±0.035
EB	1495	0.00428x+0.01347	0.99631	1.474±0.045	4.914±0.150
12DCB	1434	0.00414x+0.00982	0.99958	5.943±0.173	19.809±0.576
124TCB	1035	0.04119x+0.01743	0.99912	0.639±0.006	2.131±0.020
CAR	989	0.02025x−0.01699	0.99918	1.219±0.020	4.064±0.068
TCE	931	0.01358x−0.01522	0.97704	1.074±0.035	3.581±0.117
TeCE	910	0.07841x+0.15213	0.98604	0.560±0.014	1.866±0.047
MYR	892	0.01831x+0.06601	0.98826	2.262±0.075	7.540±0.251
pXYL	795	0.02739x−0.0645	0.97921	4.762±0.307	15.873±1.024

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
