# Peer review of "Toward the Required Detection Limits for Volatile Organic Constituents in Marine Environments with Infrared Evanescent Field Chemical Sensors"

_sensors, 2019, doi:10.3390/s19173644_

Round 1
Reviewer 1 Report
The manuscript described a mid-infrared measurement system that utilized silver halide fibers and polymer membrane to detect volatile organic compounds in seawater.
Q1. The authors should calculate the evanescent field intensity of the polymer coated fibers and justify if the sensitivity from the measurements was accurate.
Q2. The authors should provide the FTIR spectrum of the E/P-co polymer to verify the absorption of the polymer does not overlap with the analytes.
Q3. The author should provide more information regarding the polymer coating layers, such as the thickness and the uniformity. Images from SEM or optical microscope will be helpful.
Author Response
Dear Sir or Madame,
thank you for your constructive and detailed review and your valuable comments.
Q1. The authors should calculate the evanescent field intensity of the polymer coated fibers and justify if the sensitivity from the measurements was accurate.
We thank the reviewer for this comment and calculated the evanescent field intensity according to equation (4) depending on the distance x from the waveguide surface (section 2.1 Materials and reagents). At a distance equal to the penetration depth (i.e., 4.2 µm) the intensity decreased to 37% of the initial intensity (at x=0 µm, i.e., directly at the waveguide surface) at every internal reflection.
Q2. The authors should provide the FTIR spectrum of the E/P-co polymer to verify the absorption of the polymer does not overlap with the analytes.
The reference “Göbel et al., Appl. Spectrosc. 1994, 48, 678-683” was cited in section “3.1. Analysis of pollutants in seawater”. The autors illustrated a single-beam spectrum of an E/P-co polymer coated AgX fiber. The absorption feature from E/P-co overlaps with the peak from pCYM at 815 cm-1. Therefore, the peak at 1515 cm-1 was evaluated.
Q3. The author should provide more information regarding the polymer coating layers, such as the thickness and the uniformity. Images from SEM or optical microscope will be helpful.
As suggested, an image via scanning electron microscope was added to section 2.2 polymer coating. The thickness of polymer layer was calculated to be 13.7 ± 0.2 µm and was already mentioned in this section. This was calculated by differential weighing, as described by equation (5). Additionally, this information was added in the results section, i.e., “3.1. Analysis of pollutants in seawater”.
Reviewer 2 Report
Dettenrieder and coworkers presented important works about the detection of volatile organic constituents in Marine Environments. A portable sensor system was based on a cylindrical silver halide (AgX) fiber with an E/P-co coated flattened segment.
The authors introduced lots of contents in “1. Introduction”. BUT, the SENSORs reader may want to know more about the fiber sensors.
1. How about the coupling efficiency from light source to the fiber? The authors mentioned “In this study, a planar AgX fiber with cylindrical extension was used resulting in absorbance values 13-times larger than conventional cylindrical fibers [54].”. BUT, no further information about this type fiber was introduced in this work. I cannot be convinced without the introduction of the operation principles.
2. How to distinguish the detection molecules? Or how to avoid the interfere from other molecules?
In summary, the authors do not report a substantial improvement of the state of the art. The mentioned aspect of a possible replacement of other fiber sensors would have be worth a detailed investigation. However, the authors do not refer to proper the state of the art knowledge and methods which would be required for acceptance for publication.
Author Response
Dear Sir or Madame,
thank you for your constructive and detailed review and your valuable comments.
The authors introduced lots of contents in “1. Introduction”. BUT, the SENSORs reader may want to know more about the fiber sensors.
As suggested, the introduction was revised, i.e., usage of VOCs investigated within this study, commissions, and council directives were removed. A brief theory of ATR spectroscopy was added showing an image of the AgX fiber with an exemplary beam path of the IR radiation. This should aid in SENSOR readers to understand the sensing principle more clearly. Additionally, references were added showing research based on cylindrical, flattened, and flattened fiber with cylindrical extension.
How about the coupling efficiency from light source to the fiber? The authors mentioned “In this study, a planar AgX fiber with cylindrical extension was used resulting in absorbance values 13-times larger than conventional cylindrical fibers [54].”. BUT, no further information about this type fiber was introduced in this work. I cannot be convinced without the introduction of the operation principles.
Thank you very much for this comment. The operation principles were added to the introduction as requested. It is explained, that with decrease in diameter, the incident angle is changed and the penetration depth of the evanescent field and the number of internal reflections increased.
How to distinguish the detection molecules? Or how to avoid the interfere from other molecules?Molecules are detected due to their substance specific absorption features based on excitation of vibrational states as explained in the introduction. In section “3.1. Analysis of pollutants” it is explained that the specific IR bands are caused from C-H and C-Cl stretching and deformation vibrations. If other molecules interfering with the absorption bands are present, multivariate analysis, e.g., partial least square regression or principle component analysis has to be applied.
In summary, the authors do not report a substantial improvement of the state of the art. The mentioned aspect of a possible replacement of other fiber sensors would have be worth a detailed investigation. However, the authors do not refer to proper the state of the art knowledge and methods which would be required for acceptance for publication.
Research papers have been added to the introduction. We have also added the explanation that these studies showed simultaneous detection up to six different (chlorinated) hydrocarbons in water, and five VOCs in artificial seawater. These studies either used a liquid nitrogen-cooled (not applicable for in-field measurements) or a Stirling-cooled detector. The developed sensor system herein is miniaturized in size, and is suitable for in-field measurements. We were able to determine ten VOCs in real seawater simultaneously, and achieved detection limits in the low ppm to ppb regime.
We agree with the relevance of explaining the method/measurement principle, and have added this to the introduction.
Round 2
Reviewer 1 Report
Recommend for publication.